# Prompt architecture induces methodological artifacts in large language models

Melanie Brucks◉*◔, Olivier Toubia◔

Columbia Business School, New York, NY, United States of America

◔ These authors contributed equally to this work.
* mb4598@columbia.edu

## Abstract

We examine how the seemingly arbitrary way a prompt is posed, which we term "prompt architecture," influences responses provided by large language models (LLMs). Five large-scale, full-factorial experiments performing standard (zero-shot) similarity evaluation tasks using GPT-3, GPT-4, and Llama 3.1 document how several features of prompt architecture (order, label, framing, and justification) interact to produce methodological artifacts, a form of statistical bias. We find robust evidence that these four elements unduly affect responses across all models, and although we observe differences between GPT-3 and GPT-4, the changes are not necessarily for the better. Specifically, LLMs demonstrate both response-order bias and label bias, and framing and justification moderate these biases. We then test different strategies intended to reduce methodological artifacts. Specifying to the LLM that the order and labels of items have been randomized does not alleviate either response-order or label bias, and the use of uncommon labels reduces (but does not eliminate) label bias but exacerbates response-order bias in GPT-4 (and does not reduce either bias in Llama 3.1). By contrast, aggregating across prompts generated using a full factorial design eliminates response-order and label bias. Overall, these findings highlight the inherent fallibility of any individual prompt when using LLMs, as any prompt contains characteristics that may subtly interact with a multitude of hidden associations embedded in rich language data.

## Introduction

Generative artificial intelligence (AI) models and tools, in particular large language models (LLMs) such as GPT (generative pre-trained transformer), are rapidly disrupting many industries and fields of study, with many scientists and practitioners actively contemplating the use of LLMs as a substitute for humans in a wide range of tasks and occupations [1–11]. At first glance, LLMs seem particularly well-suited for some of the more tedious tasks typically performed by humans, such as combing through

**Data availability statement:** All data and code used to analyze the data are available on OSF, https://osf.io/2pzh9, DOI: 10.17605/OSF.IO/2PZH9

**Funding:** The author(s) received no specific funding for this work.

**Competing interests:** The authors have declared that no competing interests exist.

hordes of textual data and evaluating these data on various dimensions. LLMs have extensive memory and processing abilities, are presumably not influenced by emotions or moods, and never lack motivation.

As generative AI continues to gain traction, however, researchers are cautioning against hasty and indiscriminate use because these models tend to perpetuate pernicious social stereotypes and prejudices that may be embedded in the content on which they are trained. For example, due to co-occurrences in their corpora, LLMs may reveal gender stereotypes, such as perceiving a "nurse" more as a "woman" and a "doctor" more as a "man" [12–14]. As such, important ongoing work has begun developing tools for detecting and addressing this type of bias in the content created by LLMs [15–17].

In this article, we complement work that defines bias as "the predisposition or inclination toward a particular group (based on gender or race), often rooted in social stereotypes, that can unduly influence the representation or treatment of particular groups" [16] by focusing on *methodological artifacts*, or biases due to the specificities of a given task design. Methodological artifacts are a form of statistical bias, as they are systematic errors that lead to conclusions that are partly "a result of the particular research technique employed, rather than an accurate representation of the world" [18]. Using a choice architecture framework, we examine how arbitrary associations embedded in the rich language data used to train LLMs may induce statistical bias (and, therefore, errors) in output.

We propose that prompt architecture biases LLM evaluations similarly to how the context of a choice biases human judgment [19–23]. We define "prompt architecture" as the seemingly arbitrary way a prompt is posed, such as the order of options in a prompt and the framing of a prompt. For example, managers (researchers) interested in replacing their workforce (human subjects) with generative AI may ask GPT to evaluate the similarity between different sets of items. They also need to decide how to label each set in the prompts they submit to GPT, the order in which to list the sets, and how to word the exact task. For example, a user may decide to write the prompt: "Is Set B or Set C closer to Set A?" Evaluative tasks such as these, with multiple choices that are ordered and labeled, span many different LLM use cases, including diagnoses [9,11], opinion polls and surveys [2,3], and social science experiments [5,7,8].

When only one prompt is used in these evaluation tasks, the implicit assumption is that GPT operates under the normative principle of procedure invariance, revealing a stable and reliable measure of similarity between options regardless of response order, labeling, framing, or justification (i.e., asking for reasons). In support of this assumption, GPT claims "my answer will only be driven by the actual items in each set. The order or labels of the sets will not influence my answer" (S1 Fig). Contrary to this claim, we find that the arbitrary architecture of the prompt (i.e., order, label, framing, and justification) systematically and significantly biases the output of LLMs, rendering the results from LLMs prone to methodological artifacts [18]. In other words, without taking this form of bias into account, a user may erroneously conclude that set A is closer to set B, candidate 1 is more qualified than candidate 2, or diagnosis I is more likely than diagnosis II, simply because of the architecture of the prompt.

Contemporaneous work in computer science and medicine offers preliminary support of our proposition that the architecture of the prompt may affect LLM responses. For example, research finds that the order of options in multiple-choice questions affects LLMs' answers [24–27] and that this bias can be exacerbated when the task becomes more difficult [24]. Research also finds that the responses provided by LLMs are subject to the labels given to multiple-choice options [28–30]. Moreover, nascent work documents that prompt wording can influence LLM judgment. For example, Arroyo et al. [31] find that the performance of LLMs on tasks such as mortality prediction vary based on slight differences in the wording of the instructions.

We present a unifying theory of prompt architecture that can explain this work and expand on it, by framing it within a broader type of methodological artifacts to which LLMs are vulnerable. To test this, we design a full factorial experiment to systematically examine and disentangle the impact of multiple elements of prompt architecture. Specifically, in addition to order and labeling [32], we test two other classic choice architectural elements demonstrated in psychology—framing [33] and justification [34,35]—and examine how these elements interact with order and label bias within the same prompt. Mirroring findings in choice architecture, we find robust evidence that order, label, framing, and justification affect LLM responses. In particular, we uncover response-order and label bias and demonstrate how framing and justification affect these biases. We also test strategies to mitigate the methodological artifacts elicited by prompt architecture. We find that the use of uncommon labels reduces (but does not eliminate) label bias but exacerbates response-order bias for GPT-4 (it does not reduce either bias in Llama 3.1) and that specifying to the LLM that the order and labels of the items have been randomized does not alleviate either response-order or label bias in GPT-4 or Llama 3.1. By contrast, aggregating across prompt architectures generated using a full factorial design eliminates response-order and label bias. Although developing a full factorial design each time a user prompts a LLM may be impractical, the benefits likely outweigh the costs for deci- sions where bias is particularly harmful, such as in hiring or diagnoses. In addition, such factorial design may be imple- mented automatically (and potentially in a way that is transparent to the end user) by developing simple applications that generate the design and run it on the API.

By demonstrating the effect of different prompt architecture elements on LLM output, we hope to reorganize the emerg- ing literature which has narrowly focused on the effect of specific elements of order or labeling under the umbrella of prompt architecture. Specifically, choice architecture argues that "there is no such thing as a neutral design" [19]. Similarly, there is no such thing as a neutral prompt: inherent in any individual prompt is an arbitrary architecture that includes order, labeling, framing, and a host of other characteristics that may subtly impact LLM output. As a result, "prompt engineering," which typically attempts to identify one optimal prompt for a given task, may be a futile exercise. We suggest that rather than attempting to remove bias, users aggregate across different prompts, canceling out random and unpredictable noise due to the architecture of any individual prompt [36].

## Study 1

### Materials and methods

Our main test uses the application programming interface (API) of GPT-4 to perform a large number of evaluations. For replicability and because the behavior of GPT-4 keeps evolving [37], we pinned GPT-4 to its 06/13/2023 version (accessed 02/12/2024; the 06/13 version was the latest pinned version that was not in "preview" mode as of January 2024). We set temperature to 0 to obtain GPT-4's most probable answer to each prompt. Borrowing from a standard psychology task [38], we show GPT-4 three sets of items (e.g., three sets of five countries) and ask whether the second or third set is closer to the first. This type of task is described as zero-shot because we do not provide any training exam- ple in the prompt. For a given triplet {Set1, Set2, Set3}, our experimental design has 32 conditions, in a 2^5 full factorial design (Table 1 for a summary of the conditions, Fig 1 for an example of the task, and S1 File for more detail).

First, we vary whether the prompt describes the sets using letters (A, B, C) or symbols (#, %, *). We examined both letter and symbol labels to test whether letters produce additional label bias, as they (i) possess an inherent order (i.e., the alphabet) and (ii) could be more frequently present in the training set in ways that might create extraneous associations

**Table 1. Summary of conditions.**

| Element | Manipulation |
| --- | --- |
| Label Type | Letters (A, B, C) vs. Symbols (#, %, *) |
| Label Order | A, B, C vs. A, C, B or #, %, * vs #, *, % |
| Set Order | Set 1, Set 2, Set 3 vs. Set 1, Set 3, Set 2 |
| Framing | Which of the two sets is… "closer" vs. "farther"? |
| Justification | Please give me a precise answer… "don't explain it" vs "with an explanation" |

*Below are three sets of items in the domain of Countries. Each set A, B and C contains 5 items.*

*Set A:*
*1. Egypt*
*2. United States of America*
*3. Germany*
*4. Australia*
*5. France*

*Set B:*
*1. Brazil*
*2. Japan*
*3. South Africa*
*4. Spain*
*5. Russia*

*Set C:*
*1. Canada*
*2. United Kingdom*
*3. China*
*4. India*
*5. Argentina.*

*Which of the two sets (set B or set C) is closer to set A? Please give me a precise and short answer, don't explain it. Just answer 'Set B.' or 'Set C.'*

**Fig 1. Example prompt generated in the category of countries.** In this example sets are described using letters, ordered as A, B, C, and we ask for the closer set without an explanation.

given that letter labeling is a common practice. Second, we independently vary order by swapping the second and third labels (A, B, C vs. A, C, B or #, %, * vs. #, *, %) and the second and third sets (123 or 132). For example, the same triplet labeled with letters would be presented in four different ways (*A*. Set 1, *B*. Set 2, *C*. Set 3; *A*. Set 1, *C*. Set 2, *B*. Set 3; *A*. Set 1, *B*. Set 3, *C*. Set 2; *A*. Set 1, *C*. Set 3, *B*. Set 2). This is a full factorial design in which each set has one instance of each order and label combination, which allows us to test for statistical bias because we experimentally manipulate order and label independent of the actual content of the set. As a result, any impact of order and/or label is definitionally biased and does not reflect a real difference between sets. Third, to examine other elements of the prompt architecture, we vary the framing by asking GPT-4 either which set is *closer* to the first set or which set is *farther* away [33]. Finally, following work on justification [34,35], we vary whether we ask GPT-4 to justify its answer.

We employ stimuli sampling and generate stimuli across six different categories (e.g., countries) borrowed from prior research [16], with 30 different triplets {Set1, Set2, Set3} in each category (S2 File for more detail on the stimuli generation). In sum, our experimental design generates 5,760 observations: 6 categories × 30 triplets per category × 32 prompts per triplet. All data and code used to analyze the data are available at OSF (https://osf.io/2pzh9, https://doi.org/10.17605/OSF.IO/2PZH9).

## Results

Of the 5,760 observations, GPT-4 failed to complete the task 313 times (5.43% failure rate). We removed these observations to obtain our final dataset of 5,447 observations.

**Response-order and label bias.** Given full randomization, an unbiased responder should select the first option 50% of the time and any specific label 50% of the time. Instead, we find evidence that GPT-4 is prone to response-order and label bias. On average, across all observations, GPT-4 selected the first option in 63.21% of the cases ($p < .001$) and selected B over C in 74.27% of the cases ($p < .001$). Fig 2a and 2b break down these biases by condition. Across condition, we consistently find response-order bias and lettered-label bias. In contrast with labeling with letters, we find that label bias is reduced, but not completely eliminated, when using symbols: the symbol % is chosen over the symbol * in 54.16% of the cases (vs. 63.21% of "B" for letters, $p < .01$; Fig 2c). This finding is consistent with the explanation that label bias is due to LLMs favoring more common tokens (e.g., B over C), as this bias is reduced when using labels that

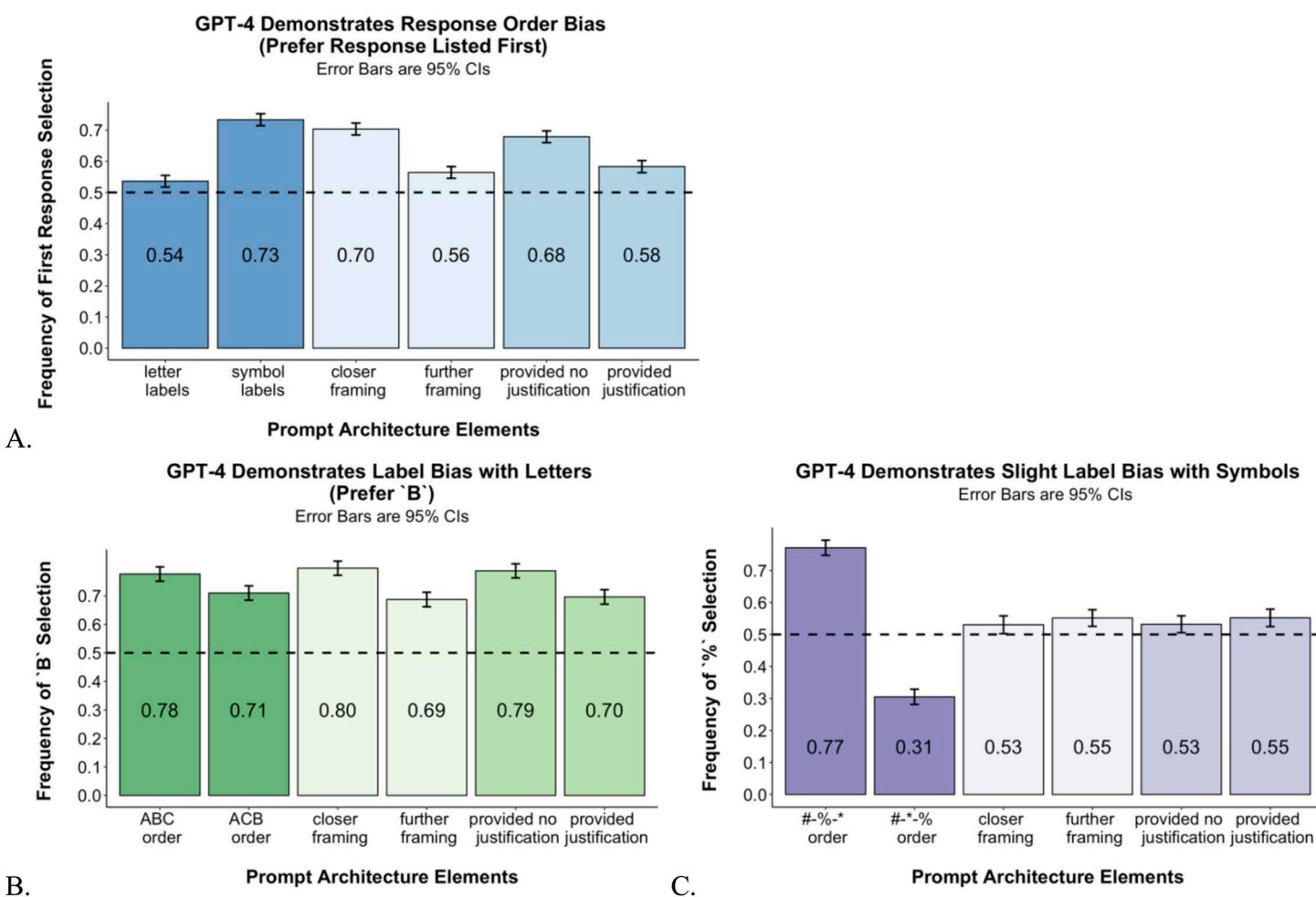

**Fig 2. Methodological bias as a result of prompt architecture. A.** Frequency of first response selection across conditions. **B.** Frequency of selecting the set labeled as B, among observations in which sets are labeled using letters. **C.** Frequency of selecting the set labeled as %, among observations in which sets are labeled using symbols. All figures show 95% confidence intervals (CIs) estimated using a random effects linear model (lmer in R), which included random intercept for triplet.

are uncommon for the task at hand. LLMs' favoring of common tokens mirrors psychological work that finds that exposure frequency positively affects people's liking [39].

**Interaction of architectural elements.** Because we systematically and jointly manipulate multiple elements of prompt architecture, we can also examine how these architectural elements interact to produce bias. First, we examine how response-order and label bias interact with each other. We find that the former is much stronger when using symbols (which are uncommon labels) than when using letters (which are common labels). GPT-4 selected the first option in 53.55% of the cases that were labeled with letters and 73.35% of the cases that were labeled with symbols. This is consistent with contemporaneous work with GPT that finds that order bias is more pronounced when options are labeled using less common letters, such as N, R, S, rather than more common letters, such as A, B, C [28]. That is, when the labels of the various options are weaker cues, GPT-4 seems to rely more heavily on other cues (i.e., the ordering of the options), suggesting that simply removing the bias from one element (e.g., labeling) will not necessarily improve LLM performance given the multifaceted nature of prompt architecture; potential bias lurks in every semantic corner of a prompt, and reducing one form of bias may simply open the opportunity for another bias to dominate.

Second, we examine the two additional architectural elements of (i) framing and (ii) justification and test whether they impact response-order or lettered-label bias. We find that asking GPT-4 for justification reduces but does not eliminate response-order bias (from the first option being selected in 67.87% of cases to 58.28% of cases) and lettered-label bias (from "B" being selected in 78.76% of cases to 69.50% of cases). This is consistent with prior work in choice architecture that finds that when people feel accountable for their decisions, they are less likely to rely on contextual cues in their decision making [35]. However, research also finds that justification can bias human decision makers in other ways. For example, asking for justification leads people to select options that contain features that are easier to articulate, even if these options are less preferred (e.g., a humorous poster vs. an abstract, aesthetic one) [34]. Although LLMs do not experience emotions or motivation, structural patterns in the training set could lead to similar results. For example, perhaps asking an LLM to justify its selection of a job candidate or image could bias it toward decisions that are easier to articulate, such as the candidate with a higher GPA or an image with more discernible components. Again, as choice architecture dictates, there is no neutral prompt. Asking for justification might bypass one type of bias but unwittingly invite another.

We also find that framing interacts with both response-order and label bias. Specifically, we find that both response-order and letter-label biases are stronger when GPT-4 is asked which set is closer to a given stimulus than when it is asked which set is farther from it (from the first option being selected in 70.36% of cases with closer framing to 56.37% of cases with farther framing, and from option "B" being selected in 79.68% of cases with closer framing to 68.60% of cases with farther framing; $ps$ <.001). Response-order and label bias are probably multiply determined, and a "closer" framing may exacerbate bias in our similarity paradigm for several reasons. One possibility could be that order and label are part of the stimuli evaluated by GPT (e.g., the content of the set is "bundled" with its letter and position, and the bundle is evaluated holistically). If this is the case, the answer could be influenced by the fact that A is closer to B than C and the first response is physically closer to the reference stimuli than the second response. Another possibility is that the "closer" framing is a more common question. The effect could then be interpreted in the context of work in psychology showing that people are more likely to rely on heuristic processing when stimuli are familiar and/or easy to process [40]. Future research could test whether a less familiar framing leads LLMs to process in a way that appears more thorough, similarly to a request for justification.

Although we do not have answers as to *why* an individual element of prompt architecture affects GPT evaluations, we provide robust evidence for the widespread and intricate effect of prompt architecture. Our results show that subtle differences in the context of choices that affect human decision making are also affecting GPT-4 evaluations. Not only did all four classic elements of choice architecture that we manipulated impact GPT-4 evaluations, but these elements interacted with each other to form a complex web of different biases. In our particular setup, the prompt that yielded the most bias was when sets were labeled A, B, C and GPT-4 was asked which set was closer (the likely default framing for many users). In this case, GPT-4 selected B (which also happens to be the first response) in a remarkable 91.67% of the cases.

We replicate these results using Llama 3.1 (S3 File) and GPT-3 (S4 File). Similar to GPT-4, Llama 3.1 exhibits response-order bias (Llama 3.1 selected the first response in 58.14% of cases) and label bias (Llama 3.1 selected B for 76.70% of the cases and % for 76.73% of the cases), and these biases are moderated by label type, framing, and justification, though not always in the same way as in GPT-4. For example, the prompt that yields the most bias is when sets are labeled #, %, * and Llama 3.1 is asked which set is closer. In this setup, Llama 3.1 selected % (which also happens to be the first response) in a remarkable 99.44% of the cases. GPT-3 also exhibits response-order bias and label bias. However, relative to GPT-4, the response-order bias was reduced (GPT-3 selected the first response in 53.16% of cases) and the label bias increased (GPT-3 selected B for 64.10% of the cases and % for 77.15% of the cases). As with GPT-4, these biases are moderated by label type, framing, and justification, though again, not always in the same way. For example, first response bias was exacerbated when GPT-3 was asked for justification (whereas it was reduced with GPT-4). These findings imply that identifying the "least biased" prompt or avoiding the "most biased" prompt may be futile, as different LLMs and different versions may react to prompt architecture differently. We examine the implications of these results further in the General Discussion.

## Study 2

Study 2 had two goals: to investigate the effect of response-order and label bias using one single-word item per set, in which evaluations are arguably much easier, and to test an intervention identified by contemporaneous research to reduce bias. Specifically, Gui and Toubia [41] find that manipulating prompt content can confound experiments due to associations in LLM training data. For example, manipulating the price in a prompt affects the kind of consumer information the LLM draws from when responding. Importantly, this work finds that specifying to the LLM that the price is randomly determined may help mitigate this confound. If the biases we observe here are due to confounding factors that correlate with the choice of order and label (e.g., more important or prominent items listed first in the training set), perhaps instructing LLM that order and label were determined randomly would similarly reduce the effect. If, however, prompt architecture effects are driven by intrinsic or idiosyncratic tendencies in favor of certain prompt characteristics, we may not observe an attenuation. We test these competing predictions in Study 2.

### Materials and methods

We again use the API of GPT-4 pinned on 06/13/203 and a temperature of 0. We employ a similar experimental design to that in our main experiment, with two modifications. First, we use sets with single items that are all single words (unigrams). Second, from the design in Study 1, we include the four conditions that orthogonally vary the order of the items and the order of the labels (A-B-C vs. A-C-B), but we only use letters as labels, only ask GPT-4 which item was closer to item A, and never ask for justification. This replicates the conditions that produced the highest level of bias in Study 1.

A crucial difference in this study is the introduction of our intervention factor in the experimental design. In the first condition (Control), we simply ask GPT-4 which of the two items is closer. In the second condition (Randomized), we add the following statement at the end of the prompt: "I have randomized the order and labels of these two items." In the third condition (Chosen), we add: "I have chosen the order and labels of these two items." If the order and label biases are due to confounding factors that correlate with the choice of order and labels, the bias may be attenuated in the Randomized condition and potentially exacerbated in the Chosen condition compared with the Control condition.

Combining these experimental factors in a 2 (item ordering) × 2 (label ordering) × 3 (Control vs. Randomized vs. Chosen) full factorial design, we have 12 prompts per triplet. We use the same six categories as in Study 1 but this time increase the number of replications to 900 per category. Therefore, our experimental design generates 64,800 observations. All data and code used to analyze the data are available at OSF (https://osf.io/2pzh9/?view_only=aa6d1b2759134b7a8408d7ccc38f1922).

## Results

Of the 64,800 observations, GPT-4 failed to complete the task 489 times (.75% failure rate). We removed these observations to obtain our final dataset of 64,311 observations.

**Response-order and label bias.** On average, across all observations, we replicate both response-order and label bias among these simpler, single-word items. GPT-4 selected the first option in 64.29% of the cases ($p < .001$) and selected response "B" in 66.72% of the cases ($p < .001$).

**Intervention.** As Fig 3a shows, order bias is consistent across the Control, Chosen, and Randomized conditions, varying only between 63.82% in the Randomized condition to 65.00% in the Chosen condition. We again find evidence of bias in favor of the label B over C, which is again consistent across conditions, as shown in Fig 3b. The frequency of the B selection varies only between 65.95% in the Randomized condition and 67.30% in the Control condition.

In summary, although in the context of using LLMs to simulate human response to price variations research finds that confounding may be reduced by informing the LLM of the source of random variation across stimuli [41], in our context, in which we ask LLMs to evaluate similarity, informing the LLM that the ordering and labeling of items have been randomized has only negligible effects and does not appear as a viable solution to the problem. Our results instead are more consistent with those of Zheng et al. [29], who find that a similar intervention does not mitigate label bias in the context of multiple-choice questions with objectively correct answers.

As in Study 1, we replicate this study design using Llama 3.1 (S5 File) and again find no evidence that the intervention worked. Instead, indicating that the order and labels were "chosen" or "randomized" flipped and exacerbated the response-order effect (among those prompts, Llama 3.1 was significantly more likely to select the *second* item). This further underscores that although the presence of prompt architecture effects is predictable, the specific impacts are unpredictable.

## Discussion

The arbitrary architecture of a prompt induces statistical bias in the responses of LLMs, with the responses partly due to "the particular research technique employed, rather than an accurate representation of the world" [18]. These insidious methodological artifacts threaten the validity of any evaluation or output that stems from a single prompt provided to an

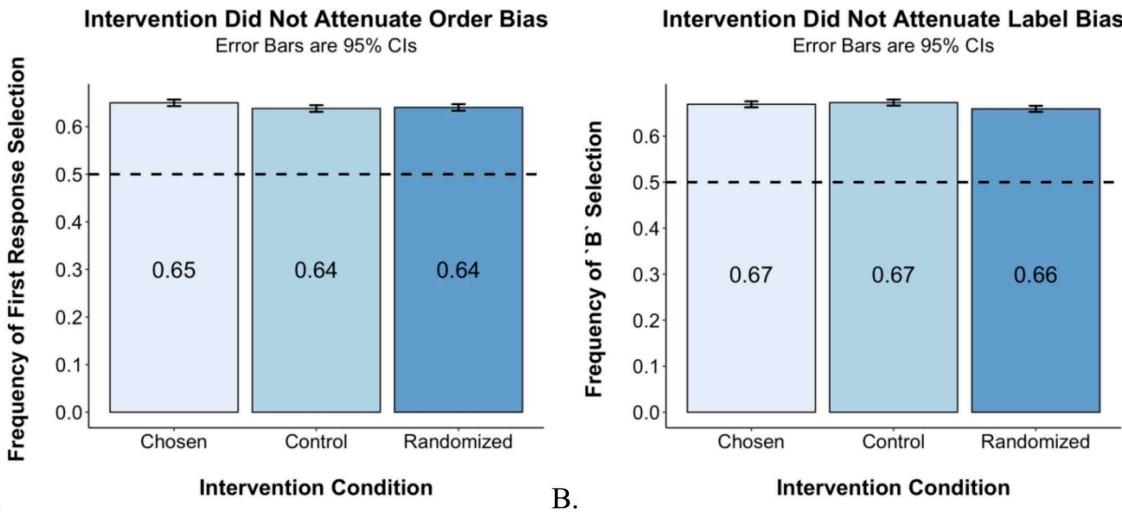

**Fig 3. Methodological bias as a result of prompt architecture (Study 2). A.** Frequency of first response selection for GPT-4 across conditions. **B.** Frequency of selecting the set labeled as B for GPT-4 among observations in which sets are labeled using letters. Both figures show 95% CIs estimated using a random effects linear model (lmer in R), which included random intercept for triplet.

LLM. For example, GPT may find that the set arbitrarily labeled as "Set B" is closer to "Set A" in as many as 80% or more of cases simply because of the architecture of the prompt. Moreover, in practice, order and labels may not be applied randomly. For example, users may list their personal favorite option first. Thus, the results we report suggest that LLMs may inadvertently serve as a confirmation bias tool rather than an unbiased adviser.

Our main goal is to present an overarching framework to understand the scope of hidden methodological biases rather than delineate the process behind any particular bias. Borrowing from extensive work in choice architecture [19], our framework suggests that there will never be such a thing as a "neutral" or "perfect" prompt. Indeed, knowing all the hidden biases embedded in rich language data is impossible; thus, specific solutions or prompt types that address bias in a certain task might not generalize to other tasks. Rather than trying to remove bias through "perfect prompts," we advise researchers and practitioners to apply the principles that human subject experiments have employed for decades—aggregating the output of multiple prompts that vary according to a full counterbalanced design, in an attempt to cancel out idiosyncratic errors of any single prompt [36].

As an initial test of this strategy for reducing bias, we examine how aggregating the output of our 32 prompts in Study 1 reduces bias in responses provided by both GPT-4 and Llama 3.1. Specifically, given our experimental design, in which we create Set 1, Set 2, and Set 3 by randomly drawing from a list of items in a category, the chance that Set 2 (vs. Set 3) is selected in a prompt asking about which is farther from or closer to Set 1 should be close to 50%. In other words, there should be nothing intrinsically different about Set 2 versus Set 3 in our 30 triplets randomly generated across six categories; thus, an unbiased responder should not show systematic preference for Set 2 over Set 3.

We find that the extent of bias varies across prompts, with the most biased prompts selecting the arbitrary Set 2 100% of the time across all triplets and categories (Fig 4). The specific prompts that demonstrate the least bias (e.g., approach 50% selection of Set 2) are different across the two models, suggesting that there is no ideal "unbiased" prompt. Instead, we suggest aggregating across different prompt types, that is, simply using the majority answer across prompt variations: for a particular triplet, across the 32 prompt formats we count the number of times the model indicated that Set 2 was closer and the number of times it indicated that Set 3 was closer, and we use the majority answer as the final answer. We find that this simple approach eliminates bias: Set 2 is selected 50% of the time when aggregating across all 32 prompts for both GPT-4 (50.01%) and Llama 3.1 (50.06%).

Note that eliminating bias does not necessarily improve accuracy. For example, a coin flip would produce an unbiased but inaccurate answer. To ensure that our intervention did not compromise accuracy, we ran an experiment identical to Study 1 but with single-word items and used Word2Vec word embeddings to calculate a noisy proxy for similarity and, in turn, GPT-4's accuracy when making similarity judgments. Our results suggest that aggregating reduces bias while increasing accuracy (S6 File). Thus, the benefit of aggregation is that it reduces bias while improving accuracy. The drawback is that it requires identifying the elements of prompt architecture that might bias an LLM's response and creating a full factorial set of prompts to address this bias, and doing so could be time and resource intensive. In cases in which this is too daunting, we recommend, at minimum, balancing order and labeling.

One concern might be whether the effects of prompt architecture are simply a snapshot in time, destined to be fixed by newer models. As an initial exploration of this possibility, we turn to our study 1 replication experiment with GPT-3. Although we observe differences between GPT-3 and GPT-4, the changes are not necessarily for the better (S4 File). Although we find significant bias in favor of the first option ($p < .001$), the bias appears to be less severe in GPT-3 than GPT-4, suggesting that, at least in terms of this particular type of bias, GPT is not improving. This is not particularly surprising given the nature of LLMs, which are trained on rich and complex natural language data. Identifying and removing any cues in a prompt's architecture seem unlikely, if not impossible, given the nature of language. In summary, the power of LLMs is undeniable, but these models should be used with discerning judgment and a thoughtful experimental design.

Our empirical context involved both simple and complex similarity tasks and formally worded prompts, which represents only a subset of the diverse types of prompts users might provide to LLMs. This raises the question of how our findings

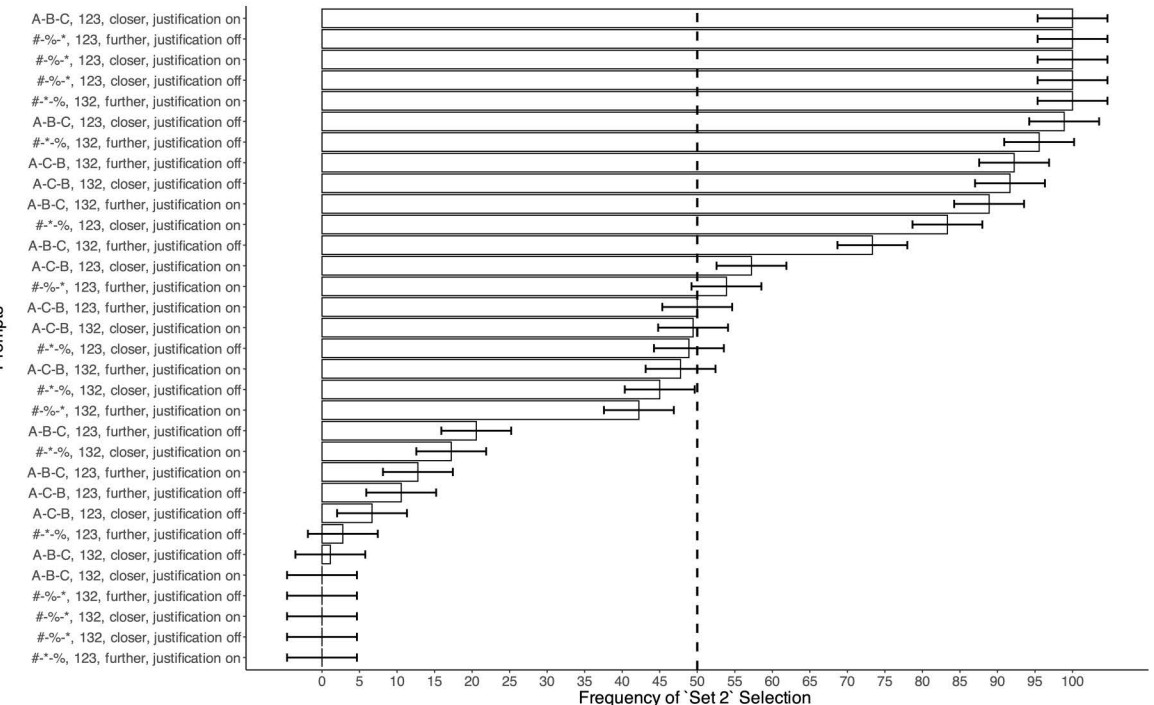

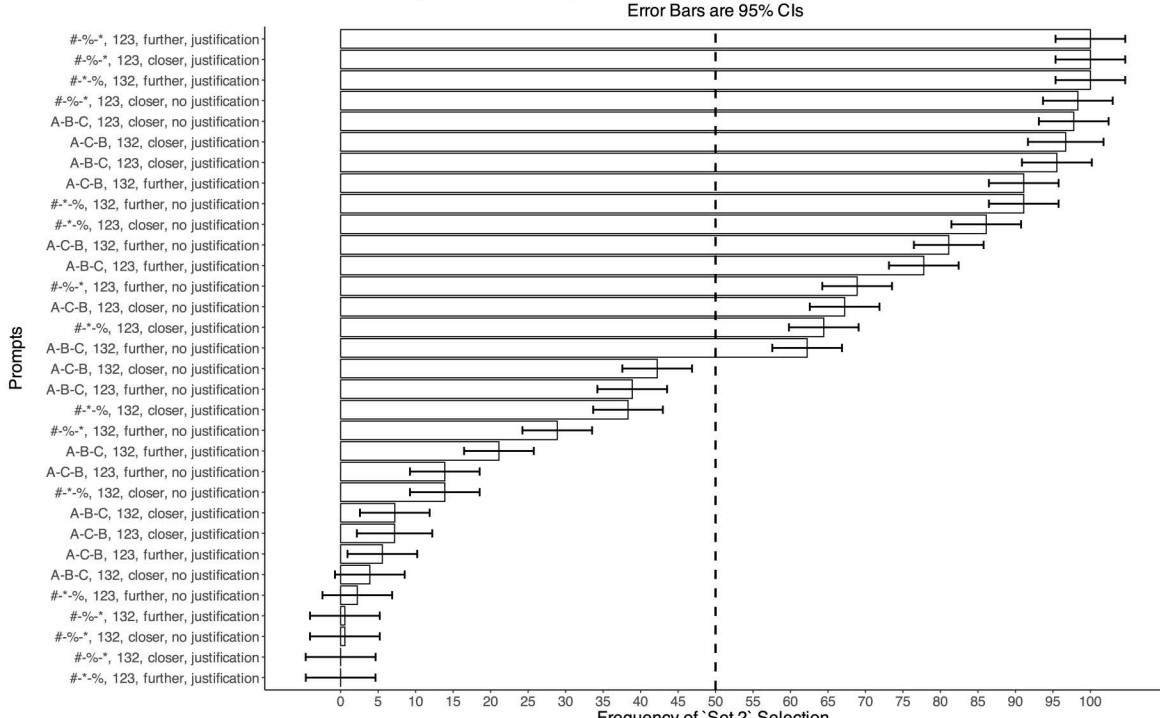

**Fig 4. Methodological bias in individual prompts.** Both figures show 95% CIs estimated using a random effects linear model (lmer in R), which included random intercept for triplet.

generalize to other contexts, such as tasks that vary in complexity beyond the similarity tasks in our studies or prompts that are phrased less formally than our prompts. Prior research with human participants provides a few hypotheses. First, human respondents are more likely to be influenced by choice architecture during complex tasks because these tasks involve more in-the-moment response construction [23,42]. Thus, our observed effects of prompt architecture may exacerbate for more complex tasks and attenuate for simpler tasks compared to the similarity task used in our studies. However, of note, we observed effects of prompt architecture for both simple (i.e., one-item stimuli) and complex (i.e., five-item stimuli) similarity tasks. Second, research suggests that informality during communication, such as text abbreviations, can signal low effort and, as a result, decrease responsiveness [43]. If this pattern exists in the training data, LLMs may be less responsive (e.g., provide less thorough justification) when the prompt contains more informal language usage. We encourage future research to explore the generalizability of our findings, including how complexity moderates the effects of prompt architecture and how additional architectural elements beyond the ones examined in this paper, such as prompt formality, may influence LLM responses.

## Supporting information

**S1 Fig. Screenshot from GPT-4 about biases.** The following screenshot was taken on 03/08/2024. We use the API Playground to mimic the conditions of our experiment: GPT-4 pinned on 06/13/23, with a temperature of 0. The "System" portion of the prompt was set to its default ("You are a helpful assistant").
(PDF)

**S1 File. Prompt variations.** Base prompt (in the categories of countries), in which sets are described using letters, ordered as A, B, C, and in which we ask for the closer set without justification.
(PDF)

**S2 File. Stimuli sampling procedure.**
(PDF)

**S3 File. Study 1: Replication using Llama 3.1.**
(PDF)

**S4 File. GPT-3 versus GPT-4.**
(PDF)

**S5 File. Study 2: Replication using Llama 3.1.**
(PDF)

**S6 File. Improving Accuracy by Using Aggregation.**
(PDF)

## Acknowledgments

We thank J. Liu and J. Zhang for help with stimuli creation and data collection.

## Author contributions

**Conceptualization:** Melanie Brucks, Olivier Toubia.

**Formal analysis:** Melanie Brucks.

**Methodology:** Melanie Brucks, Olivier Toubia.

**Visualization:** Melanie Brucks.

**Writing – original draft:** Melanie Brucks, Olivier Toubia.

**Writing – review & editing:** Melanie Brucks, Olivier Toubia.

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
