## [Decision Letter · Decision Letter 0]

30 Jul 2024

PONE-D-24-16087Prompt Architecture Induces Methodological Artifacts in Large Language ModelsPLOS ONE

Dear Dr. Brucks,

Thank you for submitting your manuscript to PLOS ONE. After careful consideration, we feel that it has merit but does not fully meet PLOS ONE’s publication criteria as it currently stands. Therefore, we invite you to submit a revised version of the manuscript that addresses the points raised during the review process.

We look forward to receiving your revised manuscript.

Kind regards,

Vijayalakshmi Kakulapati, Ph.D

Academic Editor

PLOS ONE

Journal Requirements:

3. Please upload a copy of Figure 3, to which you refer in your text on page 10. If the figure is no longer to be included as part of the submission please remove all reference to it within the text.

4. We notice that your supplementary [figures/tables] are included in the manuscript file. Please remove them and upload them with the file type 'Supporting Information'. Please ensure that each Supporting Information file has a legend listed in the manuscript after the references list.

Reviewers' comments:

Reviewer's Responses to Questions

**Comments to the Author**

1. Is the manuscript technically sound, and do the data support the conclusions?

Reviewer #1: Yes

Reviewer #2: Partly

2. Has the statistical analysis been performed appropriately and rigorously? 

Reviewer #1: Yes

Reviewer #2: Yes

3. Have the authors made all data underlying the findings in their manuscript fully available?

Reviewer #1: Yes

Reviewer #2: Yes

4. Is the manuscript presented in an intelligible fashion and written in standard English?

Reviewer #1: Yes

Reviewer #2: No

5. Review Comments to the Author

Reviewer #1: The study probes into prompt architecture—biases responses provided by Large Language Models (LLMs). Mimicking findings in choice architecture, the study finds robust evidence that label, order, framing, and asking for reasons affects LLM responses. Two major findings was, (i) the use of uncommon labels reduces (but does not eliminate) label bias but exacerbates response order bias and (ii) specifying to the LLM that the labels and order of the items have been randomized does not alleviate either response order or label bias.

Major Comments

I enjoyed reading this manuscript and believe it holds great promise. However, I have identified minor issues that require the authors' attention:

Various detailed studies in the field of Medicine have utilized multiple-choice questions, particularly in the format of A, B, and C. These studies often use complex clinical cases from sources such as the New England Journal of Medicine (NEJM) Clinicopathological Conference (CPC). With detailed reasoning, large language models (LLMs) predict the answers. These predictions, when justified, are correct answers and not biases. Any comments?

The sentence 'Although we find that asking for justification reduces response order and label bias, it is possible that it could exacerbate other biases we did not measure'. These other potential biases can be discussed further.

The sentence 'However, the bias appears to be less severe in GPT-3 compared to GPT-4, with 53.2% vs. 63.21% of the cases, respectively'. This important finding should be mentioned in the main manuscript, not only in the supplemental data.

Due to the proprietary nature of GPT models, there is growing interest in open-source LLMs. Is it possible to test the hypothesis with other open-source models like LLaMA-3?

Check the reference:

Artefacts, statistical and methodological. In: Oxford Reference.

Reviewer #2: The paper presents a highly engaging and insightful study on the biases present in GPT label generation. However, to enhance the quality and comprehensiveness of the work, I suggest the following revisions:

Firstly, the task itself needs to be more clearly defined. The authors should specify the exact types of labels they expect to encounter and analyze. This clarity will provide a better framework for understanding the scope and objectives of their study.

Secondly, the language throughout the paper requires some refinement. At certain points, the writing becomes too informal, which detracts from the scholarly tone of the work. A more formal and consistent style would improve the overall readability and professionalism of the paper.

Additionally, the authors focus exclusively on GPT-4. This focus raises concerns about potential bias towards proprietary models. It would be beneficial to include a comparison with open-source large language models (LLMs) such as LLaMA or Mistral to provide a more balanced analysis.

The definition of bias within the context of the study is another critical area that needs attention. The authors should refer to relevant definitions and frameworks, such as those discussed in the paper available at [IEEE Xplore](https://ieeexplore.ieee.org/abstract/document/10538997), to establish a definition and solid theoretical foundation for their analysis. the literature review needs to be updated to reflect the latest research and developments in the field. This update will ensure that the study is situated within the current academic discourse and acknowledges recent advancements and discussions.

Clarification is also needed on whether the labeling process employed in the study is stochastic. Understanding the nature of the labeling process is essential for assessing the reliability and consistency of the generated labels.

The authors should provide a thorough discussion of the pros and cons of their approach. This analysis will help readers understand the strengths and limitations of the method and its potential implications for future research and applications.

From what I understand, the authors aim to mitigate the biases in GPT-generated labels by proposing a more reliable approach. To enhance the practical value of their study, they should include an algorithm presented in pseudocode. This inclusion would provide a clear, actionable method for other researchers and practitioners to implement and build upon their work.

6. PLOS authors have the option to publish the peer review history of their article (what does this mean? ). If published, this will include your full peer review and any attached files.

**Do you want your identity to be public for this peer review?** For information about this choice, including consent withdrawal, please see our Privacy Policy .

Reviewer #1: **Yes: ** Balu Bhasuran

Reviewer #2: No

---

## [Author Response · Author response to Decision Letter 1]

20 Sep 2024

Response to Reviewers

We would like to thank the Academic Editor and the reviewers for the insightful and constructive comments on our manuscript, “Prompt architecture induces methodological artifacts in large language models.” We have done our best to address all of the points raised, as detailed in this letter.

Academic Editor:

Thank you for the opportunity to revise our manuscript and for providing a clear roadmap forward. Below we respond to each of your requests:

1. We have ensured that the manuscript meets the PLOS ONE style requirements.

2. All data and code are and will remain publicly available on ResearchBox: https://researchbox.org/2826&PEER_REVIEW_passcode=USUUIK. In accordance with PLOS ONE’s policy, we will make everything freely accessible if our manuscript is accepted for publication.

3. In the previous version we mislabeled Figure 3 as Figure 2. We apologize for the confusion and have corrected this typo.

4. We removed supplementary figures and tables from the main text and uploaded them separately as Supplementary Information. Following your request, each Supporting Information file has a legend listed in the manuscript after the references list.

Below, we also provide a summary of the main changes we have made in response to the reviewer’s comments:

• We have incorporated references from medicine and explained how they relate to our findings.

• We have replicated our results with an open-source model, Llama-3.1.

• We have incorporated references that study other types of biases of large language models and explained how they relate to our research.

• We have further explored the pros and cons of the aggregation approach which we propose as a way to reduce methodological artifacts.

• We have polished the writing throughout and hired a professional copy editor.

Reviewer 1:

We appreciate your insightful comments, which were essential in guiding our revision effort. Below we quote your concerns and then summarize the steps we took to address each one.

1. “Various detailed studies in the field of Medicine have utilized multiple-choice questions, particularly in the format of A, B, and C. These studies often use complex clinical cases from sources such as the New England Journal of Medicine (NEJM) Clinicopathological Conference (CPC). With detailed reasoning, large language models (LLMs) predict the answers. These predictions, when justified, are correct answers and not biases. Any comments?”

Thank you for pointing out a very important context where our findings may or may not generalize. Indeed, there are papers that find that LLMs can be quite accurate for multiple-choice questions in medicine (Harsha et al., 2023; McDuff et al., 2023; Singhal et al., 2023). However, early evidence suggests that LLM responses might also be affected by prompt architecture in the medical context. For example, Hager et al. (2024) find that the order of options in multiple-choice questions affects LLM’s answers in clinical decision making, and Arroyo et al. (2024) find that the performance of LLMs on tasks such as mortality prediction vary based on slight differences in the wording of the instructions. Therefore, we believe our findings are also relevant to the medical domain, which we now cite thanks to your input.

2. “The sentence 'Although we find that asking for justification reduces response order and label bias, it is possible that it could exacerbate other biases we did not measure'. These other potential biases can be discussed further.”

Thank you for your suggestion to further elaborate on this interesting possibility. We include a discussion of the possible biases induced by asking for justification on page 12, lines 199-204 (following “However, research also finds that asking for reasons can bias human decision makers in other ways”).

“For example, asking for justification leads people to select options that contain features that are easier to articulate, even if these options are less preferred (e.g., a humorous poster vs. an abstract, aesthetic one) (37). Perhaps asking an LLM to justify its selection of a job candidate or image could bias it toward decisions that are easier to articulate, such as the candidate with a higher GPA or an image with more discernible components.”

3. “The sentence 'However, the bias appears to be less severe in GPT-3 compared to GPT-4, with 53.2% vs. 63.21% of the cases, respectively'. This important finding should be mentioned in the main manuscript, not only in the supplemental data.”

We now mention this finding in the main manuscript on page 21, lines 370-373. We further reference the finding in the abstract.

4. “Due to the proprietary nature of GPT models, there is growing interest in open-source LLMs. Is it possible to test the hypothesis with other open-source models like LLaMA-3?”

Thank you for encouraging us to replicate our effect using an open-source LLM. Following your suggestion and R2’s, we replicated both studies using Llama 3.1. We mention these replication in the abstract and on page 13, lines 232-237 and page 16, lines 301-306 of the manuscript, and report the replications in Supplements D and F. Overall, the results support our proposition that prompt architecture induces methodological bias: we document both response-order and label effects using Llama 3.1, although GPT-4 and Llama 3.1 sometimes differ in the exact the nature of these biases (e.g., Llama 3.1 exhibits stronger symbol label bias than GPT-4). This suggests that although the presence of prompt architecture effects is robust to different models and versions, the specific impact of different features of prompt architecture is subject to change.

“We replicate these results using Llama 3.1 (see Supplemental Information D). Similar to GPT-4, Llama 3.1 exhibits response-order bias and label bias, and these biases are moderated by label type, framing, and justification, though not always in the same way as in GPT-4. Here, the prompt that yields the most bias is when sets are labeled #, %, * and Llama 3.1 is asked which set is closer. In this setup, Llama 3.1 selected % (which also happens to be the first response) in a remarkable 99.44% of the cases..”

“As in Study 1, we replicate this study design using Llama 3.1 (see Supplemental Information F) and again find no evidence that the intervention worked. Instead, indicating that the order and labels were chosen or randomized flipped and exacerbated the response-order effect (among those prompts, Llama 3.1 was significantly more likely to select the second item). This further underscores that although the presence of prompt architecture effects is predictable, the specific impacts are unpredictable.”

5. Lastly, thank you for directing our attention to an incorrect reference. We have fixed this reference. Note, it is a dictionary webpage from Oxford Reference:

Oxford Reference [Internet]. Statistical Artefact. Available from: https://www.oxfordreference.com/display/10.1093/oi/authority.20110803095426317

Reviewer 2:

We appreciate your thoughtful comments and helpful feedback. We have implemented your suggestions and believe that the paper is much stronger as a result. Below we quote your concerns and then summarize the steps we took to address each one.

1. “Firstly, the task itself needs to be more clearly defined. The authors should specify the exact types of labels they expect to encounter and analyze. This clarity will provide a better framework for understanding the scope and objectives of their study.”

Thank you for the opportunity to clarify the scope and objectives of our study. First, to clarify, labels in our context are not related to the topic of data labeling in machine learning. Instead, we are interested in the order (e.g., whether an option is presented in the first, second or third position) and labels (e.g., whether an option is labeled as “A” or “B” or “C”) assigned to the various response options in multiple choice evaluation tasks. In our context, labels are selected by the user engaging with the LLM and they should theoretically have no effect on the answer provided by the LLM because they are not a reflection of the data. We have clarified this context in the example we provide in the introduction as well as the use cases on Page 4 lines 59-65.

“For example, managers (researchers) interested in replacing their workforce (human subjects) with generative AI may ask GPT to evaluate the similarity between different sets of items. They also need to decide how to label each set in the prompts they submit to GPT, the order in which to list the sets, and how to word the exact task. For example, a user may decide to write the prompt: “Is Set B or Set C closer to Set A?” Evaluative tasks such as these, with multiple choices that are ordered and labeled, span many different LLM use cases, including diagnoses (9,11), opinion polls and surveys (2,3), and social science experiments (5,7,8).”

We also include examples of other labels that LLMs might encounter on pages 4-5, lines 75-77:

“In other words, without taking this form of bias into account, a user may erroneously conclude that set A is closer to set B, candidate 1 is more qualified than candidate 2, or diagnosis I is more likely than diagnosis II, simply because of the architecture of the prompt.”

2. “Secondly, the language throughout the paper requires some refinement. At certain points, the writing becomes too informal, which detracts from the scholarly tone of the work. A more formal and consistent style would improve the overall readability and professionalism of the paper.”

Thank you for pointing out that, at times, our writing came across as too informal. We have updated our writing throughout the paper in order to better reflect a scholarly tone. For example:

• We replaced the colloquial word “mimicking” with a more accepted term “mirroring”

• We reworded “Is it possible that a less familiar framing leads GPT to process in a way that appears more thorough, just as the request for justification?” to “Future research may test whether a less familiar framing leads LLMs to process in a way that appears more thorough, similarly to a request for justification.”

• We removed the reference suggesting that GPT might get “lazier” during holidays, as this was not from an academic source.

We also hired a professional copyeditor to further improve the paper’s writing.

3. “Additionally, the authors focus exclusively on GPT-4. This focus raises concerns about potential bias towards proprietary models. It would be beneficial to include a comparison with open-source large language models (LLMs) such as LLaMA or Mistral to provide a more balanced analysis.”

Thank you for encouraging us to replicate our effect using an open-source LLM. Following your suggestion and R1’s, we replicated both studies using Llama 3.1. We mention these replication in the abstract and on page 13, lines 232-237 and page 16, lines 301-306 of the manuscript, and, and report the replications in Supplements D and F. Overall, the results support our proposition that prompt architecture induces methodological bias: we document both response order and label effects using Llama 3.1, although GPT-4 and Llama 3.1 sometimes differ in the exact the nature of these biases (e.g., Llama 3.1 exhibits stronger symbol label bias than GPT-4). This suggests that although the presence of prompt architecture effects is robust to different models and versions, the specific impact of different features of prompt architecture is subject to change.

“We replicate these results using Llama 3.1 (see Supplemental Information D). Similar to GPT-4, Llama 3.1 exhibits response-order bias and label bias, and these biases are moderated by label type, framing, and justification, though not always in the same way as in GPT-4. Here, the prompt that yields the most bias is when sets are labeled #, %, * and Llama 3.1 is asked which set is closer. In this setup, Llama 3.1 selected % (which also happens to be the first response) in a remarkable 99.44% of the cases.”

“As in Study 1, we replicate this study design using Llama 3.1 (see Supplemental Information F) and again find no evidence that the intervention worked. Instead, indicating that the order and labels were chosen or randomized flipped and exacerbated the response-order effect (among those prompts, Llama 3.1 was significantly more likely to select the second item). This further underscores that although the presence of prompt architecture effects is predictable, the specific impacts are unpredictable.”

4. “The definition of bias within the context of the study is another critical area that needs attention. The authors should refer to relevant definitions and frameworks, such as those discussed in the paper available at [IEEE Xplore](https://ieeexplore.ieee.org/abstract/document/10538997), to establish a definition and solid theoretical foundation for their analysis. “

Thank you for encouraging us to define bias in the context of our study and for the excellent reference. We now contrast how bias has been defined and explored in prior literature with our definition to (a) better couch our findings in the extant literature and (b) clarify our contribution. In our context, bias is defined as a form of statistical bias, specifically methodological artifacts. Our work on statistical bias complements the very important extant work in LLMs on social bias, which studies “the predisposition or inclination toward a particular group (based on gender or race)... that can unduly influence the representation or treatment of particular groups.” (Raza et al., 2024) These edits are reflected in the text on pages 3-4, lines 47-55:

“In this article, we complement work that defines bias as “the predisposition or inclination toward a particular group (based on gender or race), often rooted in social stereotypes, that can unduly influence the representation or treatment of particular groups” (16) by focusing on methodological artifacts, or biases due to the specificities of a given task design. Methodological artifacts are a form of statistical bias, as they are systematic errors that lead to conclusions that are partly “a result of the particular research technique employed, rather than an accurate representation of the world” (18). Using a choice architecture framework, we examine how arbitrary associations embedded in the rich language data used to train LLMs may induce statistical bias (and, therefore, errors) in output.”

Defining our bias as a form of statistical bias provides a foundation for our design and analysis, as we delineate on page 7, lines 134-138:

“This is a full factorial design in which each set has one instance of each order and label combination, which allows us to test for statistical bias because we experimentally manipulate order and label independent of the actual content of the set. As a result, any impact of order and/or label is definitionally biased and does not reflect a real difference between sets.”

We believe the updated manuscript provides a fuller picture of how statistical bias can manifest in LLM output, clarifies our contribution, and grounds our work by providing a theoretical foundation for analysis.

5. “The literature review needs to be updated to reflect the latest research and developments in the field. This update will ensure that the study is situated within the current academic discourse and acknowledges recent advancements and discussions.”

Thank you for ensuring our literature review of this fast-moving field is up to date. We have updated the status of our references, surveyed recent additions to the academic discourse and identified new papers to include in our review.

In particular, we added the following references that have studies LLM biases of a social nature:

• Raza S, Bamgbose O, Chatrath V, Ghuge S, Sidyakin Y, Muaad AY. Unlocking bias detection: leveraging transformer-based models for content analysis [Internet]. arXiv; 2024 [cited 2024]. Available from: http://arxiv.org/abs/2310.00347

• Raza S, Rahman M, Zhang MR. BEADs: bias evaluation across domains [Internet]. arXiv; 2024 [cited 2024]. Available from: http://arxiv.org/abs/2406.04220

• Zack T, Lehman E, Suzgun M, Ro

---

## [Decision Letter · Decision Letter 1]

24 Nov 2024

PONE-D-24-16087R1

Prompt architecture induces methodological artifacts in large language models

PLOS ONE

Dear Dr. Brucks,

Thank you for submitting your manuscript to PLOS ONE. After careful consideration, we feel that it has merit but does not fully meet PLOS ONE’s publication criteria as it currently stands. Therefore, we invite you to submit a revised version of the manuscript that addresses the points raised during the review process.

We look forward to receiving your revised manuscript.

Kind regards,

Vijayalakshmi Kakulapati, Ph.D

Academic Editor

PLOS ONE

Journal Requirements:

Reviewers' comments:

Reviewer's Responses to Questions

**Comments to the Author**

1. If the authors have adequately addressed your comments raised in a previous round of review and you feel that this manuscript is now acceptable for publication, you may indicate that here to bypass the “Comments to the Author” section, enter your conflict of interest statement in the “Confidential to Editor” section, and submit your "Accept" recommendation.

Reviewer #1: All comments have been addressed

Reviewer #3: (No Response)

2. Is the manuscript technically sound, and do the data support the conclusions?

Reviewer #1: Yes

Reviewer #3: Partly

3. Has the statistical analysis been performed appropriately and rigorously? 

Reviewer #1: Yes

Reviewer #3: Yes

4. Have the authors made all data underlying the findings in their manuscript fully available?

Reviewer #1: Yes

Reviewer #3: Yes

5. Is the manuscript presented in an intelligible fashion and written in standard English?

Reviewer #1: Yes

Reviewer #3: Yes

6. Review Comments to the Author

Reviewer #1: The authors have thoroughly addressed all my comments. The manuscript is now suitable for acceptance in its current form.

Reviewer #3: Thank you for the opportunity to review this manuscript following the first round of revisions. The authors have clearly invested significant effort in addressing earlier feedback, and the manuscript is notably stronger as a result. The study’s relevance and its multidisciplinary approach are commendable, as it bridges theories between psychology, decision science, and artificial intelligence. However, a few remaining questions and suggestions are detailed in the attached document. I look forward to reading future installments of this work.

7. PLOS authors have the option to publish the peer review history of their article (what does this mean? ). If published, this will include your full peer review and any attached files.

**Do you want your identity to be public for this peer review?** For information about this choice, including consent withdrawal, please see our Privacy Policy .

Reviewer #1: **Yes: ** Balu Bhasuran

Reviewer #3: No

---

## [Author Response · Author response to Decision Letter 2]

3 Jan 2025

We included our response to reviewers as an attached document.

---

## [Decision Letter · Decision Letter 2]

29 Jan 2025

Prompt architecture induces methodological artifacts in large language models

PONE-D-24-16087R2

Dear Dr. Brucks,

We’re pleased to inform you that your manuscript has been judged scientifically suitable for publication and will be formally accepted for publication once it meets all outstanding technical requirements.

Kind regards,

Vijayalakshmi Kakulapati, Ph.D

Academic Editor

PLOS ONE

Reviewers' comments:

Reviewer #3: Thank you for the opportunity to review the newly revised manuscript. The authors have successfully addressed all of my comments, resulting in a more comprehensive manuscript. I have no further questions and wish the authors the best of luck with this project.

---

## [Editor Report · Acceptance letter]

PONE-D-24-16087R2

PLOS ONE

Dear Dr. Brucks,

I'm pleased to inform you that your manuscript has been deemed suitable for publication in PLOS ONE. Congratulations! Your manuscript is now being handed over to our production team.

Kind regards,

on behalf of

Dr. Vijayalakshmi Kakulapati

Academic Editor

PLOS ONE